# Clustering Paraphrases for Substitutability

## Abstract

Current state-of-the-art models for *lexical substitution* – the task of nominating substitutes for a word in context – ignore word sense, instead relying on powerful vector and embedded word representations to find good substitutes. We present a simple method for improving the lexical substitution rankings of existing models by integrating word sense inventories, filtering substitutes from the correct sense to the top of the rankings. To enable maximum coverage of our method, we also propose a novel method for clustering paraphrases by word sense with substitutability in mind. Our method results in sense clusters that are more substitutable and have wider coverage than existing sense inventories. They can be applied as a filter over lexical substitution rankings generated by existing vector- and embedding-based ranking models to significantly improve their performance.

## 1 Introduction

Paraphrases are alternate ways of expressing the same meaning. Natural language processing applications like machine translation (Denkowski and Lavie, 2010) and semantic representation (Yu and Dredze, 2014; Faruqui et al., 2015) benefit from large paraphrase resources that enable them to identify different words and phrases with equivalent meaning. To meet this need, there have been several efforts to automatically acquire lexical and phrasal paraphrases. The largest of these is the Paraphrase Database (PPDB) (Pavlick et al., 2015), which contains over 20M English paraphrase pairs.

Since words and phrases can be polysemous, their paraphrases can be divided into subsets representing the different meanings. For example,

paraphrases of the noun *paper* include *sheet, page, notepaper, newspaper, daily, publication, press* which can be grouped into two sense clusters $c_1$: {sheet, page, notepaper} and $c_2$: {journal, daily, publication, press}, analogous to WordNet synsets (Miller, 1995). Earlier work (Apidianaki et al., 2014; Cocos and Callison-Burch, 2016) proposed automatically clustering paraphrases by sense, rather than manually specifying sense clusters as WordNet does, in order to organize large paraphrase resources by sense.

There is a clear relationship between the sense of a word or phrase in the context of some sentence, and our ability to replace it with one or more of its paraphrases. For instance, in the sentence *The local **papers** took photographs of the footprint*, the paraphrases from $c_2$ would be good substitutes that preserve the overall meaning of the sentence, while paraphrases from $c_1$ would be poor substitutes because they change the meaning.

The task of automatically replacing a word with its meaning-equivalent paraphrases in context is called *lexical substitution* (lexsub). It was initially conceived as a practical alternative evaluation for Word Sense Disambiguation (WSD) systems (McCarthy and Navigli, 2007). Accordingly, many early lexsub models approached the problem in two steps: first, identify the sense of the target word by reference to an existing sense inventory (e.g. WordNet), and second, propose substitutes corresponding to the predicted sense (Hassan et al., 2007; Yuret, 2007). But current state-of-the-art lexical substitution systems ignore the sense disambiguation step altogether, instead relying on vector space models to find substitutes that are semantically similar to the target, and contextually appropriate (Melamud et al., 2015; Roller and Erk, 2016; Apidianaki, 2016).

Our hypothesis is that a high-coverage, *substitutable* sense inventory – one in which words belonging to a single sense are mutually substitutable

| Sentence | Annotated Substitutes (Count) | Applicable WordNet+ sense |
|---|---|---|
| Tasha snatched it from him to rip away the **paper**. | sheet (3), page (3), note (1), notepaper (1), parchment (1) | $c_1$: sheet, piece of paper, sheet of paper |
| An hour later, the **paper**'s circulation manager, Miguel Soler, was shot and killed near his home. | newspaper (4), daily (1), publication (1), press (1) | $c_2$: press, daily, gazette, tabloid |
| The operators at the East Fishkill factory wear light nylon uniforms, light blue shoe coverings and translucent hair nets made of **paper**. | sheet (1), paper pulp (1), parchment (1), paper stock (1), card stock (1) | $c_3$: material, stuff, substance |

Table 1: Example annotated sentences for the target *paper.N* from the CoInCo dataset and the corresponding WordNet+ senses (see sec. 2.1). Numbers after each substitute indicate the number of annotators who made that suggestion.

in most contexts – can be used in combination with a word-based lexsub model to improve the quality of proposed substitutes. In this paper we propose a novel method for clustering paraphrases in PPDB which improves performance on the lexical substitution task. We use a multi-view clustering algorithm that incorporates different views of paraphrase similarity – such as lexical substitution rankings, overlap in foreign translations, and similarity to WordNet synsets – and which can be tailored specifically for the task of lexical substitution. It is generalizable to all parts of speech and languages.

Our method results in sense clusters that are more substitutable than existing sense inventories as measured by their agreement with human-generated lexsub annotations. They can be applied as a filter over lexsub rankings generated by existing vector- and embedding-based ranking models to significantly improve lexsub performance.

## 2 Sense Inventories and their Substitutability

A crucial question when evaluating a sense inventory for its fitness for the lexsub task is, how well does it match human judgments about the interchangeability of words? We propose a cluster quality metric, B-Cubed F-Score (Amigó et al., 2009), as a way of comparing an automatically-generated sense inventory against a human-generated set of substitutability judgements.

We can describe a sense inventory, $C_t$, as a set of sets, $\{c_1, c_2, \ldots c_k\}$, where each subset contains synonyms of the target $t$ corresponding to one of its senses. For example, the first sense of *paper* in Table 1 is the set $c_1 = \{sheet, piece\_of\_paper, sheet\_of\_paper\}$.

A (human-annotated) lexsub dataset contains multiple sentences for each target word, and a set of annotated substitutes for each sentence (Table 1). We can view the annotated substitutions as a clustering, $L_t = \{l_1, l_2, \ldots l_3\}$, where each cluster contains the proposed substitutes for a single sentence. In Table 1, for example, the cluster $l_1$ contains $\{sheet, page, note, notepaper, parchment\}$.

We call $E_t$ the set of all $t$'s synonyms that appear in both its sense inventory and its lexsub dataset. We then use the extended B-Cubed F-score cluster evaluation metric to quantify the level of agreement between the gold standard sense labels and the proposed substitute clusters for words $e \in E_t$. B-Cubed F-score ($F_\beta^{B^3}$) measures cluster quality in terms of B-Cubed precision (measuring the extent to which each sentence's annotated substitutes come from the same gold standard sense(s)), and B-Cubed recall (measuring the extent to which each sentence's annotated substitutes cover an entire sense). Full details of our B-Cubed F-Score implementation are in the supplemental material.

We define the *substitutability* of $t$'s sense inventory, $C_t$, with respect to its annotated substitutes, $L_t$, as $F_\beta^{B^3}(E_t)$. Given many target words for which we have senses and annotations, we can average the substitutability of an entire sense inventory $C$ over its targets $T$ (for which $|E_t| \geq 2$) with annotations in $L$ into a single substitutability score:

$$subst(C, L) = Avg_{t \in T}(F_\beta^{B^3}(E_t)) \quad (1)$$

### 2.1 Measuring Substitutability

We now assess the substitutability of the following existing sense inventories with respect to human-

| Sense Inventory | $|T|$ | Avg Synonyms per Tgt | Avg Senses per Tgt | Avg CoInCo Overlap | CoInCo $B^3F_{0.5}$ |
|---|---|---|---|---|---|
| WordNet+ | 155K | 29.7 | 4.5 | 5.5 | 0.841 |
| TWSI | 1K | 16.7 | 2.42 | 6.0 | 0.842 |
| PPDBClus | 1.7M | 132.8 | 13.7 | 16.6 | 0.736 |

Table 2: Characteristics and substitutability ($B^3F_{0.5}$) of each sense inventory analyzed in our study. The Avg CoInCo Overlap indicates the average number of words appearing in both the sense inventory and CoInCo annotations, by target word.

annotated judgements. This will establish a score that we aim to outperform with our new method for generating paraphrase sense clusters.

**WordNet+.** The first sense inventory we evaluate, WordNet+, is formed from WordNet 3.0. For each CoInCo target word that appears in WordNet, we take its senses to be its synsets, with lemmas belonging to hypernyms and hyponyms of each synset included.

**PPDBClus.** We also evaluate a much larger, abeit noisier, sense inventory of words in the Paraphrase Database (PPDB) XXL version (Cocos and Callison-Burch, 2016). This sense inventory was automatically generated via spectral clustering, with PPDB2.0 Score serving as the similarity metric. The resulting sense inventory has more than four times the coverage of WordNet, but is also quite noisy.

**TWSI.** The Turk Bootstrap Word Sense Inventory v2 (TWSI) (Biemann, 2013) is a crowd-sourced sense inventory for nouns. It was developed specifically with substitutability in mind using an iterative, bootstrapped process for word sense induction and lexical coverage.

The size and characteristics of each sense inventory are given in Table 2.

For each sense inventory, we evaluate its B-Cubed F-Score over all sentences in our Concepts In Context (CoInCo) (Kremer et al., 2014) test set (see sec. 4.1). We ignore targets for which $|E_t| < 2$. We find that the average substitutability scores for WordNet and TWSI are significantly higher than that of PPDBClus when aggregated over the entire CoInCo dataset.

## 3 Multi-View Clustering for Substitutability

We propose a novel method for automatically generating a sense inventory that is tailored to a substitutability metric. Our method produces a sense inventory that has greater coverage than WordNet, and that results in much better lexical substitutions than previous automatically sense-clustered paraphrases (Cocos and Callison-Burch, 2016).

Our method uses a multi-view clustering algorithm to automatically create sense clusters of paraphrases for a given target word. Different 'views', or representations, allow us to encode properties of paraphrases that are related to our goal of substitutability, including:

- Substitutability of the paraphrase for the target word, over a large number of sentences (Context Substitutability View),

- Strength of the paraphrase's relationship with other paraphrases of the target word (Paraphrase Similarity View),

- Translation probability between the paraphrase and foreign words across multiple languages (Shared Translations View), and

- Affinity between the paraphrase, and WordNet synsets (WordNet Synsets View)

### 3.1 Multi-View Clustering

In order to cluster the paraphrases of a target word, we use the multi-view non-negative matrix factorization model of Liu et al. (2013). Nonnegative matrix factorization (NMF) approaches cluster an input matrix by finding two smaller matrices that approximately equal the input when multiplied together. If the input matrix is $X \in \mathbb{R}^{M \times N}$, where the $N$ columns represent paraphrases to be clustered and the $M$ rows represent features, NMF finds non-negative matrices $U \in \mathbb{R}^{M \times K}$ and $V \in \mathbb{R}^{N \times K}$ such that their product is approximately $X$; $UV^T \approx X$. $K$ indicates the number of clusters.

Multi-view NMF is an adaptation of NMF that jointly factorizes $n$ input matrices $\{X^1, X^2, \ldots X^n\}$, each having the same number of columns to be clustered (but an arbitrary number of rows, or features). Each $X^i$ gives a different *view* or representation of the data. Multi-view NMF assumes that the views are

complementary, and that each view is likely to cluster the data the same way on its own. Under this assumption, Multi-view NMF works by factorizing each $X^i$ as above, but adds an additional regularization component that coaxes each resulting coefficient matrix, $V^i$, toward a common consensus matrix $V^*$. Full details of our multi-view NMF implementation are in the supplemental material.

Using multi-view NMF allows us to incorporate multiple types of information about paraphrases into the clustering algorithm. We now give an overview of the views used in our experiments.

### 3.1.1 View 1: Context Substitutability

Our ultimate goal is to find clusters of paraphrases that are mutually substitutable in context. The most direct way to do this is to represent each paraphrase in terms of its 'fit' within a large number of contexts. Intuitively, paraphrases that fit well within the same subset of sentences should be clustered into the same sense.

The question of how to measure the 'fit' of a paraphrase in context remains. One option might be to use existing human-annotated lexsub data, or to crowdsource a new set of human annotations for each target word. But we want to create a method that is scalable and generalizable. Therefore, we choose to measure lexical substitutability using a simple but high-performing vector space model, AddCos (Melamud et al., 2015).

The AddCos method quantifies the fit of substitute word for target word in context by measuring the semantic similarity of the substitute to the target, and the similarity of the substitute to the context, by comparing word and context embeddings generated by the *skip-gram with negative sampling* model (Mikolov et al., 2013b,a). Full details on our implementation of the AddCos metric are in the supplemental material.

The first view that we provide as input to the multi-view NMF algorithm is a *paraphrase-sentence* matrix defined as $X^S \in \mathbb{R}^{M^S \times N}$, where $N$ denotes the number of paraphrases to be clustered, and $M^S$ is an arbitrarily large number (we use 1000 in our experiments). Each row in $X^S$ corresponds to a sentence containing the target word that we extract randomly from AGiga. Each entry $x_{ij}$ gives the AddCos score for paraphrase $j$ and target $t$ in context $W_i$: $x_{ij} = AddCos(p_j, t, W_i)$.

### 3.1.2 View 2: Paraphrase Similarity

PPDB contains a measure of paraphrase strength called the PPDB2.0 Score. It is a supervised metric designed to align with human judgements of paraphrase quality (Pavlick et al., 2015). If the paraphrases for a target pertain to different senses, then we can expect paraphrases within a given sense to be connected with high PPDB2.0 Scores. Previous work validated this idea, using PPDB2.0 Score as the basis for clustering paraphrases by sense (Cocos and Callison-Burch, 2016).

The third view used in our clustering is a *paraphrase-paraphrase* matrix $X^P \in \mathbb{R}^{N \times N}$, where $N$ is the number of paraphrases to be clustered. Each entry $x_{i,j}$ contains the PPDB2.0 Score between paraphrases $p_i$ and $p_j$. The PPDB2.0 Score is not symmetric, so we use the maximum score between paraphrases $p_i$ and $p_j$ as the entry for both $x_{ij}$ and $x_{ji}$.

### 3.1.3 View 3: WordNet Synsets

We also experiment with incorporating the clean structure of WordNet to encourage words belonging to the same synset, and others similar to them, to be clustered into the same sense.

The last view that we provide as input to multi-view NMF is defined as the *paraphrase-synset* matrix $X^W \in \mathbb{R}^{M^W \times N}$, where $N$ again gives the number of paraphrases for the target, and $M^W$ gives the number of WordNet synsets that the target belongs to. Each entry in the matrix, $x_{ij}$ gives the cosine similarity between the word embedding for paraphrase $p_j$ and a compositional synset embedding for synset $c_i$. To compute synset embeddings, we take the weighted average of the word embeddings for each lemma in the synset, where each lemma embedding is weighted by the maximum of its PPDB2.0 Score with the target word and 1:

$$v_c = \frac{\sum_{l \in c} max(1, PPDB2.0Score(l, t)) \cdot v_l}{\sum_{l \in c} max(1, PPDB2.0Score(l, t))}$$

### 3.1.4 View 4: Shared Translations

Foreign translations can also be used as a proxy for word sense (see, for example, **?**, **?**, **?**, McCarthy et al. (2016) and others.). This is based on the idea that if the foreign translations of a single English lemma can be partitioned into clusters, then the the English lemma is likely polysemous, and the clusters of translations represent its different senses.

PPDB contains the translation probabilities for each foreign translation of its paraphrases over

multiple languages. We use the translation probabilities to construct a *paraphrase-translation* matrix $X^F \in \mathbb{R}^{M^F \times N}$, where $N$ denotes the number of paraphrases to be clustered, and $M^F$ is the size of the set of all translations for any paraphrase of $t$. Practically this number can be very large, so we arbitrarily set a threshold translation probability of $10^{-9}$ as the minimum for a translated word's inclusion in the set. Each row in $X^F$ corresponds to a foreign word $f$. Each entry $x_{ij}$ gives the translation probability $prob(f_i|p_j)$ between the foreign word $f_i$ and the corresponding column's paraphrase, $p_j$.

## 4 Clustering Experiments

We now apply our proposed sense-clustering method to generate substitutable sense clusters within PPDB.

### 4.1 Data

We draw the target words for our clustering experiment from the Concepts In Context (CoInCo) dataset, containing over 15K sentences corresponding to nearly 4K unique target words (Kremer et al., 2014). Specifically, we use the 327 target words in CoInCo that have at least 10 example sentences. For each of the 327 eligible targets, we randomly divide the corresponding sentences into 60% training instances and 40% test instances. The resulting training and test sets have 4061 and 2091 sentences/annotations respectively. We will use the training annotations to tune a PPDB Score cutoff threshold to reduce noise, and we will use the test annotations to evaluate the substitutability of the resulting clusters.

### 4.2 Experimental Setup

For each target word in the CoInCo train and test sets, we take its paraphrases from PPDB2.0-XXL[1] with a paraphrase score of at least 2.3 as the set of words to cluster into senses. We chose 2.3 as the threshold score because it gave the best precision and recall when used as a threshold over the CoInCo training data. This reduces the coverage of our resulting sense inventory, but decreases its noise.

The multi-view clustering algorithm requires us to choose the number of clusters, $k$. For each target word to be clustered, we try clustering with a range of $k$ between 2 and 10. For each resulting

---
[1] http://paraphrase.org

clustering, we calculate the mean Silhouette Coefficient – an intrinsic measure of cluster quality that balances inter-cluster similarity and intra-cluster distance. We use the *paraphrase-paraphrase* matrix $X^P$ to as the basis for the Silhouette Coefficient calculation, as it was found to be effective for distinguishing senses in Cocos and Callison-Burch (2016). We choose as the final clustering solution, that clustering which produces the highest Silhouette Coefficient.

We experiment with the number of views used to produce the clusters. We first cluster the paraphrases using each view individually, and then combine views to see how they complement one another. Altogether we experiment with seven unique combinations of views. In the remaining discussion, we use the notation $SubstClus^X$ to denote a sense inventory derived from the view(s) included in the superscript. The superscript $C$ denotes the Context Substitutability view, $P$ denotes the Paraphrase Similarity View, $T$ denotes the Shared Translations view, and $W$ denotes the WordNet Synsets view. For example, $SubstClus^{CP}$ denotes the sense inventory resulting from clustering the Context Substitutability and Paraphrase Similarity views.

### 4.3 Optimized Clustering Results

| Sense Inventory | Avg Senses per Tgt | CoInCo $B^3F_{0.5}$ |
|---|---|---|
| $SubstClus^C$ | 2.9 | 0.774 |
| $SubstClus^P$ | 7.4 | 0.886 |
| $SubstClus^W$ | 3.2 | 0.759 |
| $SubstClus^T$ | 3.5 | 0.774 |
| $SubstClus^{CP}$ | 5.1 | 0.851 |
| $SubstClus^{CPW}$ | 5.0 | 0.843 |
| $SubstClus^{CPWT}$ | 4.9 | 0.849 |

Table 3: Characteristics and substitutability ($B^3F_{0.5}$) of each sense inventory. Each inventory has an average of 26.3 synonyms per target, and 7.0 words overlapping with CoInCo per target.

Of the resulting sense inventories, all multi-view inventories and the $SubstClus^P$ single-view inventory are more substitutable than WordNet and TWSI in terms of B-Cubed F-Score.

## 5 Sense Filtering for Lexical Substitution

We now investigate whether it is possible to improve the rankings of vector and embedding-based lexsub ranking systems by using the optimized

sense inventory as a filter. Our general approach is to take a set of ranked substitutes generated by an existing lexsub model. Then, we see whether 'filtering' the ranked substitutes to elevate words belonging to the correct sense of the target in context will improve the overall ranking results. We conduct an oracle experiment – in which we assume we know the correct sense of the target in context for a particular instance – to find the maximum possible improvement in lexsub score using this method. We also evaluate how well the sense filtering method works in practice, by using a simple WSD method to predict the correct sense of the target in context. In both cases, if sense filtering successfully improves the quality of ranked substitutes, it indicates that the sense inventory captures substitutability well.

## 5.1 Data

We use the test portion of the CoInCo subset defined in Section 4.1 to evaluate the extent to which our baseline and optimized sense inventories improve lexical substitution rankings. The test portion consists of 2091 annotated sentences for 327 target words.

## 5.2 Ranking Models

Our approach requires a set of rankings produced by a high-quality lexsub model to start. We generate substitution rankings for each target/sentence pair in the test set using two models. The first is the syntactic vector space model of Apidianaki (2016) (Syn.VSM), which demonstrated an ability to correctly choose appropriate PPDB paraphrases for a target word in context. Its vector features correspond to syntactic dependency triples extracted from the English Gigaword corpus. Syn.VSM ranks substitutes in context based on the cosine similarity of the substitute's basic vector with the target's contextualized vector. The second model we use is the AddCos model introduced in section 3.1.1, using a window of one word to either side of the target as context. Each ranking model produces a score for each (target, sentence, substitute) tuple in the test set, and ranks substitutes based on the predicted score.

## 5.3 Metrics

Lexical substitution experiments are typically evaluated using generalized average precision (GAP) (Kishida, 2005). GAP compares a set of predicted rankings to a set of gold standard rankings. GAP scores range from 0 to 1; a perfect ranking, in which all high-scoring substitutes outrank low-scoring substitutes, has a GAP score of 1.

For each sentence in the CoInCo test set, we consider the PPDB paraphrases for the target word to be the substitution candidates, and we set the CoInCo annotator frequency to be the gold score. Words in PPDB that were not suggested by annotators receive a gold score of 0.001. The predicted rankings are given by each lexsub model (Syn.VSM and AddCos).

## 5.4 Method

Sense filtering is intended to boost the lexsub rank of substitutes that belong to the most appropriate sense of the target given the context. We run our experiment as a two-step process, which is depicted in Table 4.

First, given a target and sentence from the CoInCo test set, we obtain the PPDB paraphrases for the target word and rank them using each lexsub model model. We calculate the overall *unfiltered* GAP score for each model as the average GAP over sentences in the CoInCo test set.

Next, we evaluate the ability of a sense inventory to improve the GAP score through filtering. In the oracle experiment, we find the maximum GAP score achieveable by adding a large number (10000) to the lexsub model's score for words *belonging to a single sense*. This elevates the words from the chosen sense to the top of the rankings, while preserving their relative order. If the sense inventory corresponds well to substitutability, we should expect this filtering to improve the ranking by eliminating proposed substitutes that do not fall within the correct sense cluster.

Next, having estimated the *maximum* achievable improvement in GAP score with sense filtering, we apply a simple word sense disambiguation method to see how well this method could work in practice. For each target word in context, we choose the 'best-fit' sense to be the sense most frequently represented among the top-5 ranked substitutes by the lexsub ranking model. We elevate the scores of all words in the 'best-fit' sense by adding 10000 to their ranking model score, and calculate the Best-Fit GAP score.

| Sentence | Last **year** they contributed $34 million, before tax. |
|---|---|
| Human-annotated Substitutes (score) | period (3), time (1), term (1), season (1), annum (1), annual (1) |
| AddCos Model-generated Ranked Substitutes (0.47 GAP) | month, week, summer, decade, **season**, spring, fall, semester, fiscal, autumn, winter, century, quarter, day, beginning, **period**, **time**, millennium, harvest, revenue, report, calendar, dispensation, cycle, date, **annum**, vintage, appropriation, increase, occasion, assignment, percent, edition, budget, age, sophomore, end, trend, **term**, grade, ... |
| $SubstClus^{CPWT}$ sense with maximum gold overlap (oracle) | **period**, **time**, age |
| Sense-filtered Ranked Substitutes (0.78 GAP) | *period, time, age,* month, week, summer, decade, **season**, spring, fall, semester, fiscal, autumn, winter, century, quarter, day, beginning, millennium, harvest, revenue, report, calendar, dispensation, cycle, date, **annum**, vintage, appropriation, increase, occasion, assignment, percent, edition, budget, sophomore, end, trend, **term**, grade, ... |

Table 4: Sense-filtering for lexsub. Human-annotated (gold) substitutes for the target **year** in the given sentence are ranked. The AddCos model ranks the gold words (bold text) in positions 5, 16, 17, 26, and 39. Our $SubstClus^{CPWT}$ sense with the greatest gold overlap is given (matching 2 words). We *filter* the model-generated substitutes by elevating the ranks of words from the best $SubstClus^{CPWT}$ sense, in the same relative order (italic text). The sense-filtered resulting rankings place the gold words in positions 1, 2, 8, 27, and 39, improving GAP by 0.31.

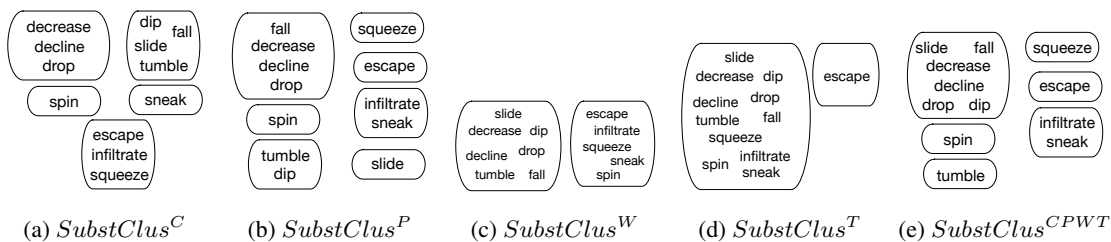

(a) $SubstClus^{C}$ (b) $SubstClus^{P}$ (c) $SubstClus^{W}$ (d) $SubstClus^{T}$ (e) $SubstClus^{CPWT}$

Figure 1: Clustering results under different views for *slip.V*

## 6 Results

We report the unfiltered, oracle sense-filtered, and best-fit sense-filtered GAP scores achieved using each of our sense inventories in Table 5.

All baseline and clustered sense inventories can improve the GAP score of the Syn.VSM and AddCos ranking models when used as a filter, based on Oracle GAP scores. This shows that filtering with a sense inventory *can* improve lexsub results. Further, in most cases, all sense inventories except PPDBClus can improve the GAP score when a simple WSD method is applied (Best-Fit GAP). Thus the sense filtering method is generally effective in practice.

Comparing the B-Cubed F-Score for the sense inventories to the Oracle GAP scores, we find that our substitutability metric is generally a good indicator of the potential for a sense inventory to improve lexsub rankings. In general, sense inventories that score highest in terms of B-Cubed F-Score also score well in terms of Oracle GAP. The Best-Fit GAP scores are not perfectly correlated with Oracle GAP scores, however, indicating that our simple WSD method may introduce bias. The simple WSD method fails to find the best sense cluster in many cases; this is an area for improvement.

Our automatically-generated sense inventories are well-suited to the task of sense filtering for lexical substitution, as indicated by their performance exceeding all baselines in terms of Oracle GAP and Best-Fit GAP when evaluated over all parts of speech. The area where our $SubstClus$ inventories are weakest is in improving the lexsub rankings for nouns. The crowdsourced TWSI significantly beats all other sense inventories in the nouns-only rankings. This suggests that performance in this task may be part-of-speech specific, and different optimization methods may be required for each part of speech.

## 7 Related Work

Early lexical substitution models performed disambiguation to propose substitutes in context (Hassan et al., 2007; Yuret, 2007). More recent works ignore explicit sense representation, instead modeling targets with contextualized vectors, against which the vectors of the substitutes are compared and ranked (Erk and Padó, 2008; Dinu and Lapata, 2010; Thater et al., 2011). Most of these works pool together the total set of substitutes available for a given target word in an anno-

| Syn.VSM Ranking Results | | | | | |
|---|---|---|---|---|---|
| | All Words | | | Nouns Only | | |
| Sense Inventory | Unfiltered GAP | Oracle GAP | Best-Fit GAP | Unfiltered GAP | Oracle GAP | Best-Fit GAP |
| $SubstClus^C$ | | 0.641 | **0.576** | | 0.647 | 0.585 |
| $SubstClus^P$ | | **0.692** | 0.540 | | 0.683 | 0.547 |
| $SubstClus^W$ | | 0.652 | 0.572 | | 0.652 | 0.581 |
| $SubstClus^T$ | | 0.654 | 0.565 | | 0.660 | 0.562 |
| $SubstClus^{CP}$ | 0.520 | 0.681 | 0.550 | 0.544 | 0.670 | 0.544 |
| $SubstClus^{CPW}$ | | 0.675 | 0.549 | | 0.669 | 0.555 |
| $SubstClus^{CPWT}$ | | 0.675 | 0.552 | | 0.669 | 0.555 |
| WordNet+ | | 0.655 | 0.563 | | 0.682 | 0.577 |
| TWSI | | – | – | | **0.756** | **0.691** |
| PPDBClus | | 0.652 | 0.432 | | 0.681 | 0.474 |

| AddCos Ranking Results | | | | | |
|---|---|---|---|---|---|
| | All Words | | | Nouns Only | | |
| Sense Inventory | Unfiltered GAP | Oracle GAP | Best-Fit GAP | Unfiltered GAP | Oracle GAP | Best-Fit GAP |
| $SubstClus^C$ | | 0.630 | **0.571** | | 0.627 | 0.568 |
| $SubstClus^P$ | | **0.686** | 0.541 | | 0.668 | 0.541 |
| $SubstClus^W$ | | 0.639 | 0.565 | | 0.634 | 0.564 |
| $SubstClus^T$ | | 0.644 | 0.560 | | 0.647 | 0.537 |
| $SubstClus^{CP}$ | 0.532 | 0.670 | 0.549 | 0.544 | 0.656 | 0.540 |
| $SubstClus^{CPW}$ | | 0.671 | 0.544 | | 0.657 | 0.540 |
| $SubstClus^{CPWT}$ | | 0.671 | 0.542 | | 0.657 | 0.539 |
| WordNet+ | | 0.651 | 0.567 | | 0.662 | 0.563 |
| TWSI | | – | – | | **0.731** | **0.687** |
| PPDBClus | | 0.648 | 0.458 | | 0.667 | 0.469 |

Table 5: Average oracle and best-fit sense-filtered GAP scores over all sentences in the CoInCo test set for each sense inventory and lexsub ranking model.

tated dataset (e.g. the SemEval-2007 Lexical Substitution dataset (McCarthy and Navigli, 2007) or the CoInCo corpus (Kremer et al., 2014)) and produce a ranking for the total substitute set. This task is called *candidate ranking*.

A more difficult, but more realistic, task is *substitute prediction* (Melamud et al., 2015) or *all-words ranking* (Roller and Erk, 2016), where the model does not have access to a gold list of candidates but needs to find possible substitutes from the entire vocabulary. This task is very challenging and the substitutes proposed by the embedding models frequently correspond to rare words. An alternative is to use existing semantic resources. Apidianaki (2016) collects substitution candidates from PPDB and ranks them using the Thater et al. (2011) models. We take this approach, nominating substitutes from a target's PPDB paraphrases.

PPDB is an enormous collection of paraphrases automatically compiled using bilingual pivoting (Bannard and Callison-Burch, 2005): the idea that two words in one language that align to the same words in a different language should be synonymous. Cocos and Callison-Burch (2016) clustered the contents of the PPDB resource by sense us-

ing a spectral clustering algorithm. The generated clusters are high coverage but contain many erroneous paraphrases, as well as paraphrases linked by different types of (non-substitutable) relations. In the substitutability-focused clustering that we propose, the resulting paraphrase clusters are more substitutable.

# 8 Conclusion

We have presented a novel method for clustering paraphrases by word sense that unites various paraphrase representations in a multi-view approach. By incorporating a view that encodes the mutual substitutability of paraphrases, our method generates paraphrase sense clusters that are more substitutable and coherent than previous results.

We have also showed that it is possible to use word senses as a filter over automatically-generated lexsub rankings to improve their agreement with human-annotated substitutes – marrying the strengths of vector- and embedding-based models with semantic information encoded in sense inventories. Our paraphrase sense clusters outperform existing sense inventories in this application when evaluated over all parts of speech.

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

# A  Supplemental Material

Here we expand on the technical details of our substitutability measurement and clustering implementation.

## A.1  Measuring Substitutability

Here we provide further details on our implementation of our substitutability metric, the extended B-Cubed F-Score (Amigó et al., 2009).

B-Cubed F-score measures cluster quality in terms of precision and recall with respect to each item being clustered. We choose this metric over other commonly-used metrics because it meets formal constraints as analyzed in Amigó et al.

(2009), and its extended version[2] allows us to assess situations where a single item belongs to multiple clusters.

Given a set of items $E$, extended B-Cubed defines the gold standard categories of element $e \in E$ as $C(e)$ and the predicted clusters of $e$ as $L(e)$. It then defines precision and recall between two items as:

$$P(e, e') = \frac{Min(|L(e) \cap L(e')|, |C(e) \cap C(e')|)}{|L(e) \cap L(e')|}$$

$$R(e, e') = \frac{Min(|L(e) \cap L(e')|, |C(e) \cap C(e')|)}{|C(e) \cap C(e')|}$$

In other words, precision indicates the extent to which two items which are clustered together, belong together. And recall indicates the extent to which two items that belong to the same gold category are clustered together.

To calculate precision ($P^{B^3}$) and recall ($R^{B^3}$) over the entire set $E$ of clustered items, B-Cubed averages the pairwise precision over all pairs of elements that are clustered together, and the pairwise recall over all pairs of elements that are in the same gold category:

$$P^{B^3}(E) = Avg_e(Avg_{e':L(e) \cap L(e') \neq \emptyset} P(e, e'))$$

$$R^{B^3}(E) = Avg_e(Avg_{e':C(e) \cap C(e') \neq \emptyset} R(e, e'))$$

Finally, we can calculate B-Cubed F-score in the usual way:

$$F^{B^3}_\beta(E) = (1 + \beta^2) \cdot \frac{P^{B^3}(E) \cdot R^{B^3}(E)}{(\beta^2 \cdot P^{B^3}(E)) + R^{B^3}(E)}$$

where $\beta$ is a parameter that corresponds to the emphasis on recall over precision. In our work we use $\beta = 0.5$ to emphasize precision over recall in the substitutability score.

To use B-Cubed F-score for measuring substitutability, we first define the set $E_t$ of words that appear in both the sense inventory for target word $t$ and the annotated substitutions for $t$. We view the annotated substitutions as a clustering, $L_t$, where $L_t(e)$ denotes the set of sentences for which word $e$ is suggested as a substitute for $t$. Finally we use the sense inventory as the set of gold standard category labels over $E_t$, such that $C_t(e)$ gives the set of senses to which $e$ belongs.

---

[2] https://github.com/hhromic/python-bcubed

We can then define the substitutability of $t$'s sense inventory, $C_t$, with respect to its annotated substitutes, $L_t$, as $F_\beta^{B^3}(E_t)$. Finally, given many target words for which we have senses and annotations, we can aggregate the substitutability of the entire sense inventory $C$ over the set of targets $T$ with annotations in $L$ into a single substitutability score:

$$subst(C, L) = Avg_t(F_\beta^{B^3}(E_t)) \qquad (2)$$

where $F_\beta^{B^3}(E_t)$ is calculated in terms of $C_t$ and $L_t$ as above.

## A.2 Multi-View Non-Negative Matrix Factorization

Given the set of PPDB paraphrases for a target word, there are several sources of information we can use to discover subsets that are mutually substitutable. Thus our setting is a good fit for multi-view clustering approaches that incorporate data from multiple *views*, or representations, of the items to be clustered.

In order to cluster the paraphrases of a target word, we use the multi-view non-negative matrix factorization model of Liu et al. (2013)[3]. Nonnegative matrix factorization (NMF) approaches cluster an input matrix by finding two smaller matrices that approximately equal the input when multiplied together. If the input matrix is $X \in \mathbb{R}^{M \times N}$, where the $N$ columns represent paraphrases to be clustered and the $M$ rows represent features, NMF finds non-negative matrices $U \in \mathbb{R}^{M \times K}$ and $V \in \mathbb{R}^{N \times K}$ such that their product is approximately $X$:

$$UV^T \approx X$$

The resulting matrices $U$ and $V$ are called the *basis* and *coefficient* matrices respectively. In the result, if $K$ represents the number of clusters of items in $X$, then the coefficient matrix $V$ provides a transformation from paraphrases in $X$ to clusters. The basis matrix $U$ also provides a mapping from features to clusters.

In order to find $U$ and $V$, NMF uses a multiplicative update method to minimize the Frobenius norm of the difference between $X$ and $UV^T$, while constraining $U$ and $V$ to be nonnegative:

$$\min_{U,V} \|X - UV^T\|_F^2, U \geq 0, V \geq 0$$

---

[3]Code available from http://jialu.cs.illinois.edu/

Multi-view NMF is an adaptation of NMF that jointly factorizes $n$ input matrices $\{X^1, X^2, \ldots X^n\}$, each having the same number of columns to be clustered (but an arbitrary number of rows, or features). Each $X^i$ gives a different *view* or representation of the data. Multi-view NMF assumes that the views are complementary, and that each view is likely to cluster the data the same way on its own. Under this assumption, Multi-view NMF works by factorizing each $X^i$ as above, but adds an additional regularization component that coaxes each resulting coefficient matrix, $V^i$, toward a common consensus matrix $V^*$. Specifically, the optimization problem becomes to minimize, over $U^i, V^i, V^*$ for $1 \leq i \leq n$:

$$\sum_{i=1}^n \|X^i - U^i V^{iT}\|_F^2 + \sum_{i=1}^n \lambda_i \|V^i - V^*\|_F^2$$
$$s.t. \forall 1 \leq k \leq K, \|U_{.,k}^{(i)}\|_1 = 1 \text{ and } U^i, V^i, V^* \geq 0$$

where each $\lambda_i$ parameter tunes the relative weight between views. In the original paper and in our experiments, we set $\lambda_i = 0.01$ for all $i$. Setting the sum of each column in the basis matrix to 1 is a normalization technique introduced in (Liu et al., 2013) that confers a probabilistic interpretation on the basis and coefficient matrices, and ensures the magnitude of each is comparable for optimization.

## A.3 AddCos Lexical Substitution Model

We use the AddCos (Melamud et al., 2015) lexsub model several places in our work. We use it first to measure the similarity between a paraphrase and a sentence in the *paraphrase-sentence* matrix used as the first view for clustering. We also use it as a ranking model for the CoInCo dataset, which we then improve upon using our sense-filtering approach. Here we provide details of our implementation.

The AddCos method quantifies the fit of substitute word $s$ for target word $t$ in context $W$ by measuring the semantic similarity of the substitute to the target, and the similarity of the substitute to the context:

$$AddCos(s, t, W)$$
$$= \frac{|W| \cdot cos(s, t) + \sum_{w \in W} cos(s, w)}{2 \cdot |W|}$$

The vectors $s$ and $t$ are word embeddings of the substitute and target generated by the *skip-gram*

*with negative sampling* model (Mikolov et al., 2013b,a). The context $W$ is the set of words appearing within a fixed-width window of the target $t$ in a sentence (we use a window of 1), and the embeddings $c$ are context embeddings generated by *skip-gram*. In our implementation, we train 300-dimensional word and context embeddings over the 4B words in the Annotated Gigaword (AGiga) corpus (Napoles et al., 2012) using the gensim word2vec package (Mikolov et al., 2013b,a; Řehůřek and Sojka, 2010). The `word2vec` training parameters we use are a context window of size 3, learning rate *alpha* from 0.025 to 0.0001, minimum word count 100, sampling parameter $1e^{-4}$, 10 negative samples per target word, and 5 training epochs.

