# Peer review of "Clustering Paraphrases for Substitutability"

_ACL 2017 — decision unknown_

[Official Review · Reviewer 1 · rating 4 · confidence 4]
soundness 3 · originality 4 · clarity 4 · impact 3 · substance 4 · appropriateness 5 · meaningful comparison 3 · presentation format Oral Presentation

This paper proposes integrating word sense inventories into existing approaches
for the lexical substitution task by using these inventories to filter
candidates. To do so, the authors first propose a metric to measure the mutual
substitutability of sense inventories with human judgments for the lexsub task,
and empirically measure the substitutability of inventories from various
sources such as WordNet and PPDB. Next, they propose clustering different
paraphrases of a word from PPDB using a multi-view clustering approach, to
automatically generate a sense inventory instead of using the aforementioned
inventories. Finally, they use these clusters with a naive (majority in top 5)
WSD technique to filter existing ranked list of substitution candidates.

- Strengths:

* The key idea of marrying vector space model based approaches and sense
inventories for the lexsub task is useful since these two techniques seem to
have complementary information, especially since the vector space models are
typically unaware of sense and polysemy.

* The oracle evaluation is interesting as it gives a clear indication of how
much gain can one expect in the best case, and while there is still a large gap
between the oracle and actual scores, we can still argue for the usefulness of
the proposed approach due to the large difference between the unfiltered GAP
and the oracle GAP.

- Weaknesses:

* I don't understand effectiveness of the multi-view clustering approach.
Almost all across the board, the paraphrase similarity view does significantly
better than other views and their combination. What, then, do we learn about
the usefulness of the other views? There is one empirical example of how the
different views help in clustering paraphrases of the word 'slip', but there is
no further analysis about how the different clustering techniques differ,
except on the task directly. Without a more detailed analysis of differences
and similarities between these views, it is hard to draw solid conclusions
about the different views.                                  

* The paper is not fully clear on a first read. Specifically, it is not
immediately clear how the sections connect to each other, reading more like
disjoint pieces of work. For instance, I did not understand the connections
between section 2.1 and section 4.3, so adding forward/backward pointer
references to sections should be useful in clearing up things. Relatedly, the
multi-view clustering section (3.1) needs editing, since the subsections seem
to be out of order, and citations seem to be missing (lines 392 and 393).

* The relatively poor performance on nouns makes me uneasy. While I can expect
TWSI to do really well due to its nature, the fact that the oracle GAP for
PPDBClus is higher than most clustering approaches is disconcerting, and I
would like to understand the gap better. This also directly contradicts the
claim that the clustering approach is generalizable to all parts of speech
(124-126), since the performance clearly isn't uniform.

- General Discussion:

The paper is mostly straightforward in terms of techniques used and
experiments. Even then, the authors show clear gains on the lexsub task by
their two-pronged approach, with potentially more to be gained by using
stronger WSD algorithms.

Some additional questions for the authors :

* Lines 221-222 : Why do you add hypernyms/hyponyms?
* Lines 367-368 : Why does X^{P} need to be symmetric?
* Lines 387-389 : The weighting scheme seems kind of arbitrary. Was this indeed
arbitrary or is this a principled choice?
* Is the high performance of SubstClus^{P} ascribable to the fact that the
number of clusters was tuned based on this view? Would tuning the number of
clusters based on other matrices affect the results and the conclusions?
* What other related tasks could this approach possibly generalize to? Or is it
only specific to lexsub?

[Official Review · Reviewer 2 · rating 2 · confidence 4]
soundness 3 · originality 4 · clarity 4 · impact 3 · substance 4 · appropriateness 4 · meaningful comparison 3 · presentation format Poster

Strengths:
The paper presents a new method that exploits word senses to improve the task
of lexical substitutability.  Results show improvements over prior methods.

Weaknesses:
As a reader of a ACL paper, I usually ask myself what important insight can I
take away from the paper, and from a big picture point of view, what does the
paper add to the fields of natural language processing and computational
linguistics.  How does the task of lexical substitutability in general and this
paper in particular help either in improving an NLP system or provide insight
about language?  I can't find a good answer answer to either question after
reading this paper.

As a practitioner who wants to improve natural language understanding system, I
am more focused on the first question -- does the lexical substitutability task
and the improved results compared to prior work presented here help any end
application?  Given the current state of high performing systems, any discrete
clustering of words (or longer utterances) often break down when compared to
continuous representations words (see all the papers that utilitize discrete
lexical semantics to achieve a task versus words' distributed representations
used as an input to the same task; e.g. machine translation, question
answering, sentiment analysis, text classification and so forth).  How do the
authors motivate work on lexical substitutability given that discrete lexical
semantic representations often don't work well?  The introduction cites a few
papers from several years back that are mostly set up in small data scenarios,
and given that this word is based on English, I don't see why one would use
this method for any task.  I would be eager to see the authors' responses to
this general question of mine.

As a minor point, to further motivate this, consider the substitutes presented
in Table 1.
1. Tasha snatched it from him to rip away the paper.
2. Tasha snatched it from him to rip away the sheet.

To me, these two sentences have varying meanings -- what if he was holding on
to a paper bag?  In that scenario, can the word "paper" be substituted by
"sheet"?  At least, in my understanding, it cannot.  Hence, there is so much
subjectivity in this task that lexical substitutes can completely alter the
semantics of the original sentence.

Minor point(s):
 - Citations in Section 3.1.4 are missing.

Addition: I have read the author response and I am sticking to my earlier
evaluation of the paper.